# Insular functional organization during handgrip in females and males with obstructive sleep apnea

**Amrita Pal**[1], **Jennifer A. Ogren**[2], **Ravi S. Aysola**[3], **Rajesh Kumar**[4,5], **Luke A. Henderson**[6], **Ronald M. Harper**[2], **Paul M. Macey**[1]*

**1** UCLA School of Nursing, University of California, Los Angeles, California, United States of America, **2** Department of Neurobiology, University of California, Los Angeles, California, United States of America, **3** Division of Pulmonary and Critical Care, David Geffen School of Medicine at UCLA, University of California, Los Angeles, California, United States of America, **4** Department of Anesthesiology, University of California, Los Angeles, California, United States of America, **5** Department of Radiological Sciences, University of California, Los Angeles, California, United States of America, **6** Department of Anatomy and Histology, Sydney Medical School, University of Sydney, Sydney, Australia

* pmacey@ucla.edu

**Data Availability Statement:** The data are available in a Harvard Dataverse public repository: https://dataverse.harvard.edu/dataset.xhtml?persistentId=doi:10.7910/DVN/LYOXJT.

## Abstract

### Study objectives

Brain regulation of autonomic function in obstructive sleep apnea (OSA) is disrupted in a sex-specific manner, including in the insula, which may contribute to several comorbidities. The insular gyri have anatomically distinct functions with respect to autonomic nervous system regulation; yet, OSA exerts little effect on the organization of insular gyral responses to sympathetic components of an autonomic challenge, the Valsalva. We further assessed neural responses of insular gyri in people with OSA to a static handgrip task, which principally involves parasympathetic withdrawal.

### Methods

We measured insular function with blood oxygen level dependent functional MRI. We studied 48 newly-diagnosed OSA (age mean±std:46.5±9 years; AHI±std:32.6±21.1 events/hour; 36 male) and 63 healthy (47.2±8.8 years;40 male) participants. Subjects performed four 16s handgrips (1 min intervals, 80% subjective maximum strength) during scanning. fMRI time trends from five insular gyri—anterior short (ASG); mid short (MSG); posterior short (PSG); anterior long (ALG); and posterior long (PLG)—were assessed for within-group responses and between-group differences with repeated measures ANOVA ($p<0.05$) in combined and separate female-male models; age and resting heart-rate (HR) influences were also assessed.

### Results

Females showed greater right anterior dominance at the ASG, but no differences emerged between OSA and controls in relation to functional organization of the insula in response to handgrip. Males showed greater left anterior dominance at the ASG, but there were also no

**Funding:** This research was supported by the National Institute of Nursing Research NR-017435 and National Institute of Heart, Lung and Blood Institute HL135562. The funders had no role in study design, data collection and analysis, decision to publish, or preparation of the manuscript.

**Competing interests:** The authors have declared that no competing interests exist.

differences between OSA and controls. The males showed a group difference between OSA and controls only in the ALG. OSA males had lower left activation at the ALG compared to control males. Responses were mostly influenced by HR and age; however, age did not impact the response for right anterior dominance in females.

## Conclusions

Insular gyri functional responses to handgrip differ in OSA vs controls in a sex-based manner, but only in laterality of one gyrus, suggesting anterior and right-side insular dominance during sympathetic activation but parasympathetic withdrawal is largely intact, despite morphologic injury to the overall structure.

## Introduction

Cardiovascular disease (CVD) in obstructive sleep apnea (OSA) is difficult to treat [1, 2], possibly due to long-term changes in the autonomic nervous system (ANS) that underpin high sympathetic tone and disrupt blood pressure regulation [3]. Associations between OSA and hypertension, coronary artery disease, stroke, arrhythmias and death are well established [4]. However, most studies exploring the comorbidities associated with OSA include both male and female participants, and evidence of whether these links are affected by sex remain under investigation [5]. A recent study of almost 2 million OSA subjects and 2 million matched controls reported that hypertension was more prevalent in women than men in the condition [6]. Furthermore, there is evidence of a female predominance of OSA-related stoke as well as more severe detrimental changes in endothelial function, peak blood flow, systemic inflammation, and digital vascular function [7, 8]. Other clinical cardiovascular characteristics, such as morning BP patterns and responses to acute BP challenges in OSA also vary by sex [3, 9]. These differences in physiologic responses between sexes in OSA presumably have an underlying neural basis.

Our earlier work has shown some sex-specific effects of OSA on neural regulation of cardiovascular stimuli. We found central autonomic responses differed in male OSA versus healthy groups [10–13]. Furthermore, although autonomic maneuvers such as the cold pressor test, hand grip, and Valsalva maneuver evoked lower amplitude, delayed onset, and slower heart rate changes in combined male and female OSA patients over healthy people [3, 14, 15], these alterations were more pronounced in females with the sleep disorder. These autonomic dysfunctions may reflect sex-related central injury. There is also evidence that hormones play a significant role in the development of OSA itself. For example, the prevalence of OSA increases dramatically following menopause in females [16], and polycystic ovary syndrome which is characterized by menstrual disturbances, excess androgen, and often obesity, is associated with increased prevalence of OSA [17]. A considerable body of literature links alterations in various hormones, including testosterone, with OSA [18], although mechanisms outlining sex-based processes in that regulation have yet to be established.

The hypothalamus is a major regulator of hormone release, and this structure in turn closely interacts with the anterior insula [19, 20]. The insula also has substantial projections to brainstem output nuclei [21]; thus, the structure is a major contributor to autonomic regulation [22, 23]. Multiple studies revealed that the insula as a whole shows significant OSA-related fMRI signal changes [15, 24–27], structural injury or adaptations [28–33], reduced perfusion [34], and altered metabolic state [35–37]. When considered across large scale networks, the

resting connectivity of the insula consistently shows OSA-related differences [32, 33, 38], suggesting either an insula-specific deficiency, or network-wide alterations. While precise regional variations in insular changes in OSA are currently being explored, the insula contains various regions with differing developmental beginnings and projection patterns. For example, the posterior insula is granular/dysgranular in nature and projects to other high-order cortical regions, whereas the anterior insular is predominantly agranular in nature and projects to lower brain regions such as the amygdala, ventral striatum, and autonomic regulatory regions, including the hypothalamus and various brainstem nuclei [19, 39]. These projection and morphologic patterns leave a potential for sex to exert differential influences on autonomic patterns in different regions.

To probe autonomic regulation, we used functional MRI (fMRI) measures of neural function during tasks that elicit an autonomic response [40]. Three standard autonomic challenges often used in combination are the cold pressor, handgrip and Valsalva maneuver; other standard tests such as tilt, Mueller maneuver, or pharmacological manipulations are less amenable to fMRI scanning [41]. The cold pressor, handgrip and Valsalva are all pressor challenges, that is they raise blood pressure, but through different mechanisms. Thus, common fMRI patterns across these challenges could be interpreted as reflecting blood pressure regulatory processes, whereas challenge-specific responses could reflect task-related effects. Both cold pressor and Valsalva challenges predominantly increase sympathetic activity during the active phase of the challenge; whereas, the handgrip adds an element of parasympathetic withdrawal in the short term (10 sec) [42–45].

In healthy adults, we showed the functional organization of the insular cortex is gyri-specific for the handgrip, a challenge that combines autonomic perturbation and an intentional motor activity [46, 47]. Specifically, the anterior insula is more activated during the early, predominantly parasympathetic withdrawal phase [48, 49]. In contrast, the strain phase of the handgrip challenge, which is associated with a moderate HR and sympathetic increase, elicited the greatest responses in the middle insular gyri. These sex-specific neural patterns co-occur with sex-specific peripheral differences, with females displaying HR smaller increases [3, 47]. The data are consistent with a recent human study showing tachycardic responses elicited by stimulating the posterior insula and bradycardic responses from stimulating the more anterior insular cortex [50]. The insula also displays a laterality effect, with the right side more closely aligned with sympathetic, and the left side parasympathetic, activity changes [48, 51, 52]. Thus, we expect insular responses to OSA during handgrip to differ by sex, hemisphere and gyral subregions [46, 53].

The objective here was to determine the nature of insular functional organization during a handgrip challenge in OSA, both controlling for sex statistically, and considering females and males separately. This study was a secondary analysis of a dataset we collected earlier. Given the anterior autonomic and left-sided parasympathetic role of the insula, and the reduced cardiovascular responses to a handgrip in OSA, we hypothesized an anterior dominance of fMRI responses in that condition, and since the left insula serves more- parasympathetic aspects, we hypothesized a greater left-side response in affected subjects. Since cardiovascular responses differ by sex in healthy people, we further hypothesized that alterations in insular organization contributed to those sex differences in people with and without OSA.

## Methods

### Participants

We studied 111 adults consisting of 48 newly-diagnosed, untreated OSA patients (36 males, 12 females) and 63 healthy control participants (43 males, 23 females); details are in Table 1.

**Table 1. Participant information.**

| | All | | | Male | | | Female | | |
|---|---|---|---|---|---|---|---|---|---|
| | CONTROL Mean ± std [Range] N = 63 | OSA Mean ± std [Range] N = 48 | *p* † OSA vs. CONTROL | CONTROL Mean ± std [Range] N = 40 | OSA Mean ± std [Range] N = 36 | *p* † OSA vs. CONTROL | CONTROL Mean ± std [Range] N = 23 | OSA Mean ± std [Range] N = 12 | *p* † OSA vs. CONTROL |
| Age (years) | 47.5 ± 8.8 [30.9–65.8] | 46.5 ± 9.0 [30.8–62.7] | 0.57 | 45.9 ± 9.1 [30.9–64.5] | 44.9 ± 8.9 [30.8–62.7] | 0.65 | 50.3 ± 7.8 [40.2–65.8] | 51.3 ± 7.9 [37.0–62.2] | 0.73 |
| BMI (m²/ kg) | 24.7 ± 3.7 [16.6–35.5] | 30.5 ± 5.1 [21.3–43.2] | *<0.001* | 25.2 ± 2.8 [17.6–29.8] | 29.6 ± 4.7 [21.4–43.2] | *<0.001* | 23.94 ± 5.0 [16.6–35.5] | 32.9 ± 5.7 [21.3–41.4] | *<0.001* |
| Resting HR (bpm) | 68.9 ± 11.6 [45.8–102.1] | 71.7 ± 9.9 [51.6–93.5] | 0.19 | 66.7 ± 11.1 [45.8–95.7] | 72.8 ± 10.4 [53.1–93.5] | *0.016* | 72.8 ± 11.5 [57.7–102.1] | 68.2 ± 7.9 [51.6–82.3] | 0.23 |
| Sleep parameters for OSA | | | | | | | | | |
| AHI (events/ hour) | n/a | 32.6 ± 21.1 [5.0–100.7] | n/a | n/a | 34.6 ± 19.7 [10.0–100.7] | n/a | n/a | 26.7 ± 24.8 [5.0–89.4] | n/a |
| SaO2 (minimum %) | n/a | 80.3 ± 9.4 [50.0–96.0] | n/a | n/a | 78.2 ± 9.4 [50.0–96.0] | n/a | n/a | 86.2 ± 6.4 [73.0–96.0] | n/a |
| SaO2 (mean%) | n/a | 94.9 ± 1.9 [88.0–97.0] | n/a | n/a | 94.9 ± 2.0 [88.0–97.0] | n/a | n/a | 94.8 ± 1.5 [92.0–97.0] | n/a |

Characteristics of OSA and control groups, with separation by sex. Group differences were tested with two-way ANOVA for OSA parameters, p values have been indicated (italicized if ≤ .05). HR was recorded in the scanner over 1 minute immediately prior to the first handgrip task. Sleep parameters were based on the patients' polysomnographic study.

† *p* for two way ANOVA F-test, group comparison OSA vs. CONTROL

Inclusion criteria for all participants included age 21–65 and weight <125kg (MRI constraint). Participants in the control group were in good health and those in the OSA group had a diagnosis of OSA. Recruitment was principally via fliers posted at the UCLA Sleep Disorders sleep clinic and on the campus and nearby communities, with additional fliers emailed or given in person upon request based on word-of-mouth. OSA patients were recruited from the UCLA Sleep Disorders Center from Dec 2005 to Aug 2008, and were recently (< 2 months) diagnosed with OSA according to the 1999 American Academy of Sleep Medicine guidelines based on an in-clinic full polysomnography test [54]. After completing a phone screening (see S1 File), participants were invited to UCLA. At the visit prior to consenting, control participants were screened for OSA using a semi-structured interview to assess daytime sleepiness, snoring, bed partner report of breathing difficulties during sleep, and nighttime gasping episodes, and referred to a full sleep study if those symptoms were present (see S1 File). Exclusion criteria for all participants included other sleep disorders, major illness or head injury, stroke, major cardiovascular disease, diabetes, current use of psychotropic or cardiovascular medications other than statins, and diagnosed mental disorder. Exclusion criteria also included MRI contraindications, including metallic implants not classified as safe at 3 Tesla, pregnancy and claustrophobia. Within the larger OSA population the present OSA sample could be considered representative of the minority of people who are relatively healthy and receiving standard healthcare (no major comorbidities, no mental health diagnoses or medications, no autonomically-active anti-hypertensive medications, limited obesity, UCLA patients).

The procedures were approved by the UCLA Institutional Review Board. All participants were provided a description of the procedures priori to visiting UCLA, and upon their visit those procedures were reviewed and participants provided written, informed consent.

## Measurements

Brain blood-oxygen level dependent (BOLD) fMRI signals were recorded in a 3.0 Tesla MRI scanner (Siemens, Magnetom, Trio) with an 8-channel head coil. We used a standard echo-planar imaging protocol (repetition time [TR] = 2000 ms; echo time [TE] = 30 ms; flip angle = 90˚; matrix size = 64 x 64; field-of-view = 230 mm x 230 mm; slice thickness = 4.5 mm). A pulse oximeter (Nonin 8600FO) with a sensor on the left index finger was used to record $O_2$ saturation and heart rate, and the plethysmographic waveform ($SaO_2$) was recorded at 1 kHz. For spatial localization, two high resolution, T1-weighted anatomical images were acquired with a magnetization prepared rapid acquisition gradient echo sequence (TR = 2200 ms; TE = 2.2 ms; inversion time = 900 ms; flip angle = 9˚; matrix size = $256 \times 256$; field-of-view = $230 \times 230$ mm; slice thickness = 1.0 mm). These two scans were realigned and averaged for each participant to result in one anatomical reference.

## Protocol

Participants were asked to refrain from coffee and other substances with stimulants for 12 hours prior to the study. While lying in a scanner, following a 1 minute baseline, participants performed four 16 s handgrips (80% subjective maximum grip strength) at 1 minute intervals against a squeeze ball without metal components, with one minute baseline after the fourth challenge. An air-filled plastic bag, connected to a pressure transducer, was placed in the participants' right hand. During the practice period, participants briefly squeezed at 100% subjective maximum strength at least two times and then at 80% for a 10–20 seconds. In the scanner, a light signal was used to indicate the onset of each grip period. Participants were instructed to squeeze to maintain the 80% pressure upon seeing the light signal. Participants practiced the static handgrip exercise maneuver prior to scanning, both outside and supine inside the MRI scanner. At least 30 min of rest (structural scanning) separated the practice from the trial periods. A pressure signal was monitored to verify that all participants performed the four static handgrip exercise tasks at the correct time. Timing was synchronized to the fMRI scans.

## Analysis: Physiology and participant characteristics

We measured HR from the $SaO_2$ plethysmographic waveform using peak detection. The median HR was calculated over the 60 second baseline period immediately prior to the first handgrip. Age, BMI, HR, and sleep parameters for the OSA group were described and compared with control groups using an ANOVA model. The full assessment of beat-to-beat HR responses was presented earlier [3].

## Analysis: MRI

We preprocessed the fMRI scans using SPM12 (https://www.fil.ion.ucl.ac.uk/spm). Images were realigned for motion correction, and linear detrended over each series. For each participant, scans were spatially normalized in two steps, first coregistering the mean fMRI to the T1 anatomical scan and then warping to the "VBM8" template in Montreal Neurological Institute (MNI) space based on the T1 "DARTEL" spatial normalization algorithm [55]. These steps resulted in all participants' fMRI images being in the template space.

The five major gyri were parcellated from the average of the high-resolution T1-weighted scans: Three short (anterior) gyri and two long (posterior) gyri: anterior short gyrus (ASG), mid short gyrus (MSG), posterior short gyrus (PSG), anterior long gyrus (ALG), and posterior long gyrus (PLG). We included these regions as mask files in nifti format (S2 File). Two experienced research team members determined the parcellation based on manual tracing with

reference to a brain atlas [56]. The regions were outlined in normalized space; although this approach is slightly less accurate than individual tracing, the resolution of the fMRI data (>50 mm$^3$) relative to the anatomical scans (< 1 mm$^3$) is such that any differences in accuracy would not be meaningful. Signal intensity changes over time were extracted from each voxel in each gyrus from the processed images. For each gyrus in each participant, a mean time trend over all voxels was then calculated. Time trends were converted to percent change relative to the mean of the 1-minute baseline period. For each participant, the signals from the four challenges were separated and averaged to create one single handgrip percent change time trend that was passed to the group level analysis. While this averaging could theoretically result in reduced sensitivity, in practice, the statistical approach we chose takes advantage of repeated measures, and could detect small effect sizes.

To assess posterior-anterior effects, signal intensity changes were calculated relative to those in the PLG. As discussed above, the importance of the anterior insula has been described in clinical and animal studies. Our previous work showed this anterior-specific role could be demonstrated by comparing the fMRI signal in the anterior vs poster-most insula (PLG), and we repeated this technique here [46]. At each time point, signal intensity changes within the PLG were subtracted from those in the ASG, MSG, PSG and ALG for each hemisphere so that direct comparisons between these regions and the PLG could be assessed. Lateralization was assessed by subtracting signal changes in each of the five left gyri from the corresponding gyri on the right side; for example, ASG laterality was calculated by subtracting the left ASG time trend from the right ASG time trend.

The resulting fMRI signals were assessed for within- and between-group differences using repeated measures ANOVA (RMANOVA). The analysis was implemented with SAS "proc mixed", as described earlier [57, 58]. In brief, this approach assesses within-group changes and between-group differences over time, with each 2 sec time-point during and after the challenge assessed relative to baseline time-points. We applied the Tukey-Fisher criterion for multiple comparisons; that is, we assessed the overall model for significance ($p \leq 0.05$), and then effects of interest (time, group by time), before considering individual time-points of difference. The latter tests are performed within the "proc mixed" procedure, as the output includes time-point tests of significance (hence no post-hoc tests were needed). We assessed the effects in combined male-female models with sex as a covariate and in sex-specific models.

The RMANOVA mixed model approach allows for continuous variables to be included, so we performed secondary analyses of age and resting HR. We created four models that included different age effects added to the main model (group + time + group x time):

1. Main + age: age effects independent of group over the entire protocol, independent of time;

2. Main + age + age x group: group-specific effects of age over the entire protocol, independent of time;

3. Main + age + age x time: age effects on handgrip responses, independent of group.

4. Main + age + age x group + age x time + age x group x time: age effects on between-group differences in handgrip responses.

We repeated these calculations for models with HR in place of age. For the purposes of this study, we only focused on the within- and between-OSA and control group responses in the different models. The age-by-time and HR-by-time measures are not independent of the main effects of interest, but the degree to which these secondary models affect the within and between-group p-values reflects potential associations between the clinical and fMRI measures.

## Heart Rate During 4 Handgrip Tasks

**Fig 1. HR physiological changes moment to moment.** The HR (beats per minute) were illustrated with the mean (smooth lines) and stdev (faded lines) for control (grey) and OSA (black) groups in "all" (both sexes), males and females. The four handgrip challenges, along with the baseline and recovery during the experiment have been illustrated here. The first 60 seconds baseline denoted by the box was averaged to get the resting HR reported.

## Results

### Participants

Table 1 shows participant characteristics. Age was similar between OSA and control groups, and as expected, BMI was high in OSA over control. Unlike females, resting HR was higher in male OSA compared with male control participants.

### Resting and evoked HR responses

The HR response during the baseline, handgrip and recovery for the 4 challenges have been shown previously [3]. Control and OSA groups displayed significant increases in HR during each of the 4 handgrip periods (Fig 1). Splitting by sex, male OSA subjects had significantly greater baseline HR compared with male controls, while female OSA subjects displayed

**Table 2. Intrinsic PLG changes in the left and right insula.**

| p-values & model statistics for RMANOVA | Model details | All | | Male | | Female | |
|---|---|---|---|---|---|---|---|
| | | Left PLG | Right PLG | Left PLG | Right PLG | Left PLG | Right PLG |
| Main model group, time group, time | ChiSq (p value) | 64.1 (<0.001) | 52.35 (<0.001) | 82.84 (<0.001) | 62.78 (<0.001) | 3.68 (0.30) | 2.6 (0.45) |
| | Fit (−2 log-likelihood) | 23525.1 | 23750.3 | 16054.5 | 16214.9 | 7330 | 7390.3 |
| | Group effect p-value (mean over entire series for each group) | 0.73 | 0.71 | 0.92 | 0.87 | N/A | N/A |
| Handgrip response: within group | Time (within-group effect of time) p-values | | | | | | |
| | **Main** | **0.49** | **0.48** | **0.31** | **0.11** | **N/A** | **N/A** |
| | Age | 0.49 | 0.48 | 0.31 | 0.11 | N/A | N/A |
| | Age x Group | 0.49 | 0.48 | 0.31 | 0.11 | N/A | N/A |
| | Age x Time | 0.81 | 0.81 | 0.49 | 0.52 | N/A | N/A |
| | Age x Group x Time | 0.81 | 0.81 | 0.49 | 0.52 | N/A | N/A |
| | HR | 0.49 | 0.48 | 0.31 | 0.11 | N/A | N/A |
| | HR x Group | 0.49 | 0.48 | 0.31 | 0.11 | N/A | N/A |
| | HR x Time | 0.36 | 0.45 | 0.53 | 0.64 | N/A | N/A |
| | HR x Group x Time | 0.36 | 0.45 | 0.53 | 0.64 | N/A | N/A |
| Handgrip response: between-group | Time X Group (between-group effect of time) p-values | | | | | | |
| | **Main** | **0.51** | **0.54** | **0.28** | **0.32** | **N/A** | **N/A** |
| | Age | 0.51 | 0.54 | 0.28 | 0.32 | N/A | N/A |
| | Age x Group | 0.51 | 0.54 | 0.28 | 0.32 | N/A | N/A |
| | Age x Time | 0.50 | 0.52 | 0.28 | 0.32 | N/A | N/A |
| | Age x Group x Time | 0.50 | 0.52 | 0.28 | 0.32 | N/A | N/A |
| | HR | 0.51 | 0.54 | 0.28 | 0.32 | N/A | N/A |
| | HR x Group | 0.51 | 0.54 | 0.28 | 0.32 | N/A | N/A |
| | HR x Time | 0.66 | 0.69 | 0.60 | 0.64 | N/A | N/A |
| | HR x Group x Time | 0.66 | 0.69 | 0.60 | 0.64 | N/A | N/A |

Salient statistics and p-values from 9 RMANOVA models for left and right PLG in three sets (mixed, male, female). Full data are available online [59]. The main model (bold) is the interaction of group-by-time (fMRI = group + time + group x time), and statics of significance and fit are in the top rows of the table. The "Group" effect is the mean over the entire series and does not represent responses, and is not discussed. The two effects of interest "Time", which represents within-group responses over time, and "Time x Group", which represents between-group differences in responses. The p-values for these effects are shown for the 9 models. All models include the main effects plus additional mean or interaction terms. All interaction models also include means. For example, "Age x Time" is fMRI = group + time + group x time + age + age x time.

reduced baseline HR levels. Furthermore, while male OSA subjects displayed robust increases in HR during each handgrip period, female OSA subjects show few such HR changes.

## Intrinsic PLG changes in left and right insula

Table 2 and Fig 2 show the left and right insula intrinsic PLG responses. The overall model was not significant for females (model statistics in Table 2). For males, the model was significant, with a small decrease in signal intensity during each handgrip task. However, there were no within- or between-group effects of the handgrip task (Table 2).

## Laterality

Fig 3 shows the laterality of the insula responses during the handgrip, with positive values reflecting right dominance. Table 3 shows the model statistics. The right-minus-left percentage

Intrinsic changes in PLG: fMRI Handgrip Responses of Left and Right Insula PLG

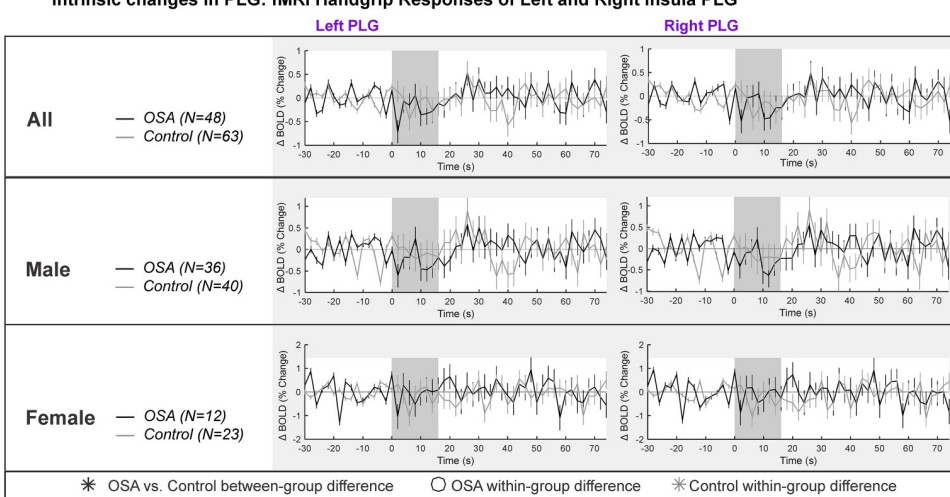

**Fig 2. PLG intrinsic changes in left and right insula.** Left and right hemisphere fMRI signals of PLG are demonstrated for combined (top panel), males (middle panel) and females (bottom panel). The graphs reflect baseline (group mean ± SE), averaged over challenges, with time-points of significant increase or decrease relative to baseline within-group, and time-points of between-group differences (RMANOVA, $p < 0.05$).

change fMRI signal for each gyrus is shown for all participants, and separately for males and females. In the combined participants, and in the females there was no group difference in the response of the right compared with the left during the handgrip for any insula gyri. That is, the overall model statistics in Table 3 show that no between-group differences appeared in the combined groups and in females. However, males showed a significant difference in OSA vs control for the ALG insular gyrus. At the ASG anterior gyrus, there were significant right activation effects by 0.2% in females, whereas males showed a significant left activation by 0.2% at the initial 8s of sympathetic activation phase for handgrip. However, these opposite lateralization effects were not significantly different in OSA vs controls. The influence of age or HR on the main effects was noted where *p*-values changed substantially. These covariates did modify the time it took for the subject to respond to the handgrip task.

## Left side anterior-posterior organization

Fig 4 represents the left insula anterior-to-posterior functional organization during handgrip, with positive values representing greater anterior dominance. Table 4 shows the model statistics. The mean±SEM percentage change fMRI signal for each of left ASG, MSG, PSG and ALG with respect to PLG is shown for all participants, and separately for males and females. The signal intensity in the left ASG changes during handgrip were approximately 0.5% higher relative to those changes in the left PLG in both control and OSA groups for both males and females. The overall model statistics in Table 4 show that no between-group differences between controls and OSA appeared in the combined, males or females groups for the left ASG, MSG, PSG or ALG. The males and females showed sympathetic activation effects in the initial sympathetic phase of handgrip at the left insula. The influence of age or HR on the main effects was noted where *p*-values changed substantially.

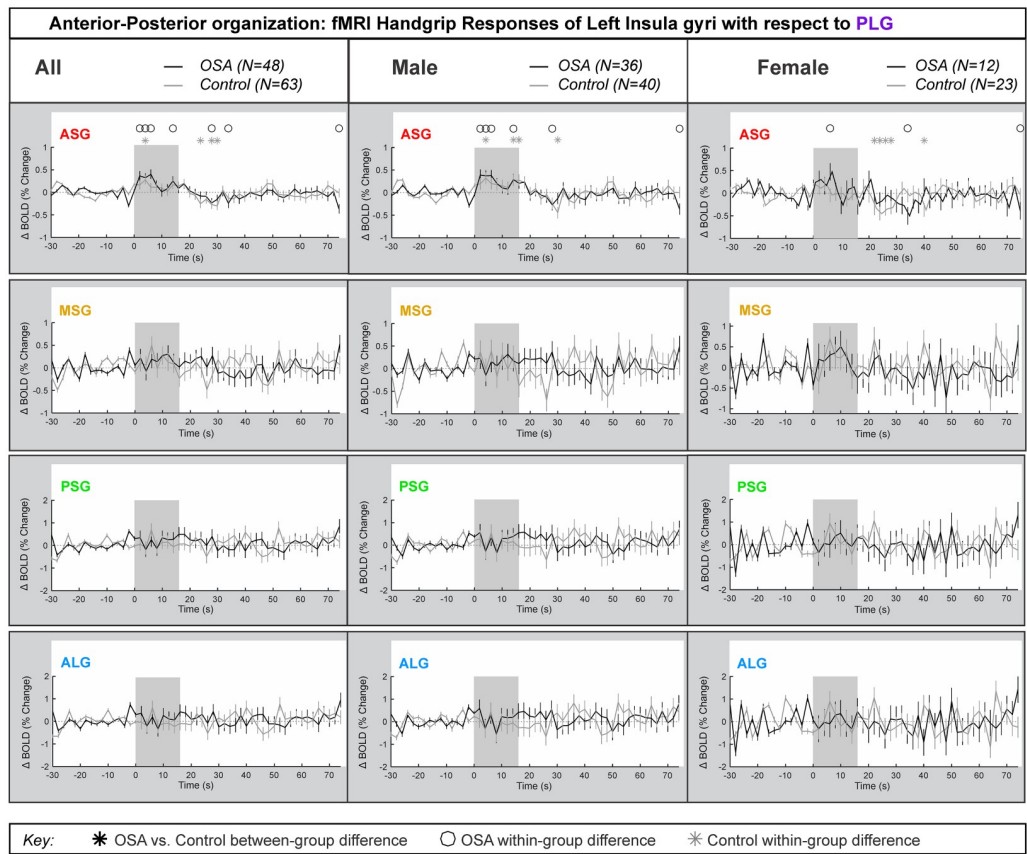

**Fig 3. Lateralization by gyri.** Right hemisphere fMRI signals relative to left hemisphere for all gyri (group mean ± SE), averaged over challenges, with time-points of significant increase or decrease relative to baseline within-group, and time-points of between-group differences (RMANOVA, p < 0.05) for the right-left laterality insular gyri responses in participants (all in left, male in middle and females in right columns respectively).

## Right side anterior-posterior organization

Fig 5 represents the right insula anterior-to-posterior functional organization during handgrip, with positive values representing greater anterior dominance. Table 5 shows the model statistics. The percentage change fMRI signal for each of right ASG, MSG, PSG and ALG with respect to right PLG is shown for all participants, and separately for males and females. Similar to the left side, while signal intensity during a handgrip was greater in right ASG (by 0.4% in females and by 0.2% in males), relative to the right PLG, there were no significant differences between the control and OSA groups. The other three gyri did not respond significantly to the handgrip task. Table 5 shows that no between-group differences appeared in the combined, male or female groups. However, unlike the left insula, age did not impact the right insula's anterior dominance of ASG during the handgrip task in females. Although in males, the interaction of age by time (meaning age influences on handgrip responses) affected all within-group effects at the ASG anterior gyrus, but age had no influence on any between-group effects. The interaction of resting HR and time (meaning HR influence on handgrip responses) influenced within-group effects for females and males in the ASG.

The full model results, including averaged timetrends as presented in the Figs, are available in a data repository [59].

**Table 3. Laterality of insular fMRI organization.**

| Model details | All ASG | All MSG | All PSG | All ALG | All PLG | Male ASG | Male MSG | Male PSG | Male ALG | Male PLG | Female ASG | Female MSG | Female PSG | Female ALG | Female PLG |
|---|---|---|---|---|---|---|---|---|---|---|---|---|---|---|---|
| **Main model: group, time** | | | | | | | | | | | | | | | |
| ChiSq ($p$ value) | 227.1 (<0.001) | 83.1 (<0.001) | 180.6 (<0.001) | 121.1 (<0.001) | 113.7 (<0.001) | 193.5 (<0.001) | 82.2 (<0.001) | 152.7 (<0.001) | 95.0 (<0.001) | 111.6 (<0.001) | 16.7 (<0.001) | 10.6 (0.01) | 22.8 (<0.001) | 31.5 (<0.001) | 7.0 (0.07) |
| Fit (−2 log-likelihood) | 5461.1 | 15181.7 | 10712.8 | 7682.9 | 9458.7 | 4077.9 | 10590.3 | 7704 | 5541.2 | 6691.8 | 1389.7 | 4569.9 | 3012.6 | 2204 | 2834.4 |
| Group effect p-value (mean over entire series for each group) | 0.63 | 0.78 | 0.47 | 0.09 | 0.93 | 0.82 | 0.93 | 0.46 | 0.013 | 0.87 | 0.07 | 0.46 | 0.37 | 0.16 | N/A |
| **Time (within-group effect of time) p-values** | | | | | | | | | | | | | | | |
| **Handgrip response: within-group** | | | | | | | | | | | | | | | |
| **Main** | **<0.001** | **0.65** | **0.67** | **<0.001** | **<0.01** | **<0.001** | **0.52** | **0.50** | **0.003** | **0.1** | **<0.01** | **0.93** | **0.45** | **0.22** | **N/A** |
| Age | <0.001 | 0.65 | 0.67 | <0.001 | <0.01 | <0.001 | 0.52 | 0.50 | 0.003 | 0.1 | <0.01 | 0.93 | 0.45 | 0.22 | N/A |
| Age x Group | <0.001 | 0.65 | 0.67 | <0.001 | <0.01 | <0.001 | 0.52 | 0.50 | 0.003 | 0.1 | <0.01 | 0.93 | 0.45 | 0.22 | N/A |
| Age x Time | 0.53 | 0.31 | 0.09 | 0.47 | 0.81 | 0.55 | 0.048 | 0.12 | 0.15 | 0.72 | 0.07 | 0.39 | 0.30 | 0.06 | N/A |
| Age x Group x Time | 0.53 | 0.31 | 0.09 | 0.47 | 0.81 | 0.55 | 0.048 | 0.12 | 0.15 | 0.72 | 0.07 | 0.39 | 0.30 | 0.06 | N/A |
| HR | <0.001 | 0.65 | 0.67 | <0.001 | <0.01 | <0.001 | 0.52 | 0.50 | 0.003 | 0.1 | <0.01 | 0.93 | 0.45 | 0.22 | N/A |
| HR x Group | <0.001 | 0.65 | 0.67 | <0.001 | <0.01 | <0.001 | 0.52 | 0.50 | 0.003 | 0.1 | <0.01 | 0.93 | 0.45 | 0.22 | N/A |
| HR x Time | 0.41 | 0.22 | 0.21 | 0.35 | 0.39 | 0.34 | 0.13 | 0.60 | 0.37 | 0.75 | 0.66 | 0.89 | 0.72 | 0.94 | N/A |
| HR x Group x Time | 0.41 | 0.22 | 0.21 | 0.35 | 0.39 | 0.34 | 0.13 | 0.60 | 0.37 | 0.75 | 0.66 | 0.89 | 0.72 | 0.94 | N/A |
| **Time X Group (between-group effect of time) p-values** | | | | | | | | | | | | | | | |
| **Handgrip response: between-group** | | | | | | | | | | | | | | | |
| **Main** | **0.60** | **0.51** | **0.64** | **0.12** | **0.39** | **0.21** | **0.20** | **0.96** | **0.016** | **0.85** | **0.78** | **0.93** | **0.66** | **0.83** | **N/A** |
| Age | 0.60 | 0.51 | 0.64 | 0.12 | 0.39 | 0.21 | 0.20 | 0.96 | 0.016 | 0.85 | 0.78 | 0.93 | 0.66 | 0.83 | N/A |
| Age x Group | 0.60 | 0.51 | 0.64 | 0.12 | 0.39 | 0.21 | 0.20 | 0.96 | 0.016 | 0.85 | 0.78 | 0.93 | 0.66 | 0.83 | N/A |
| Age x Time | 0.62 | 0.50 | 0.58 | 0.13 | 0.36 | 0.23 | 0.22 | 0.94 | 0.015 | 0.83 | 0.76 | 0.94 | 0.64 | 0.78 | N/A |
| Age x Group x Time | 0.62 | 0.50 | 0.58 | 0.13 | 0.36 | 0.23 | 0.22 | 0.94 | 0.015 | 0.83 | 0.76 | 0.94 | 0.64 | 0.78 | N/A |
| HR | 0.60 | 0.51 | 0.64 | 0.12 | 0.39 | 0.21 | 0.20 | 0.96 | 0.016 | 0.85 | 0.78 | 0.93 | 0.66 | 0.83 | N/A |
| HR x Group | 0.60 | 0.51 | 0.64 | 0.12 | 0.39 | 0.21 | 0.20 | 0.96 | 0.016 | 0.85 | 0.78 | 0.93 | 0.66 | 0.83 | N/A |
| HR x Time | 0.83 | 0.71 | 0.71 | 0.19 | 0.50 | 0.64 | 0.50 | 0.99 | 0.10 | 0.94 | 0.60 | 0.87 | 0.46 | 0.86 | N/A |
| HR x Group x Time | 0.83 | 0.71 | 0.71 | 0.19 | 0.50 | 0.64 | 0.50 | 0.99 | 0.10 | 0.94 | 0.60 | 0.87 | 0.46 | 0.86 | N/A |

Salient statistics and p-values from 9 RMANOVA models for right-minus-left insula organization in three sets (mixed, male, female). Full data are available online [59]. The main model (bold) is the interaction of group-by-time (fMRI = group + time + group x time), and statics of significance and fit are in the top rows of the table. The "Group" effect is the mean over the entire series and does not represent responses, and is not discussed. The two effects of interest "Time", which represents within-group responses over time, and "Time x Group", which represents between-group differences in responses. The p-values for these effects are shown for the 9 models. All models include the main effects plus additional mean or interaction terms. All interaction models also include means. For example, "Age x Time" is fMRI = group + time + group x time + age + age x time.

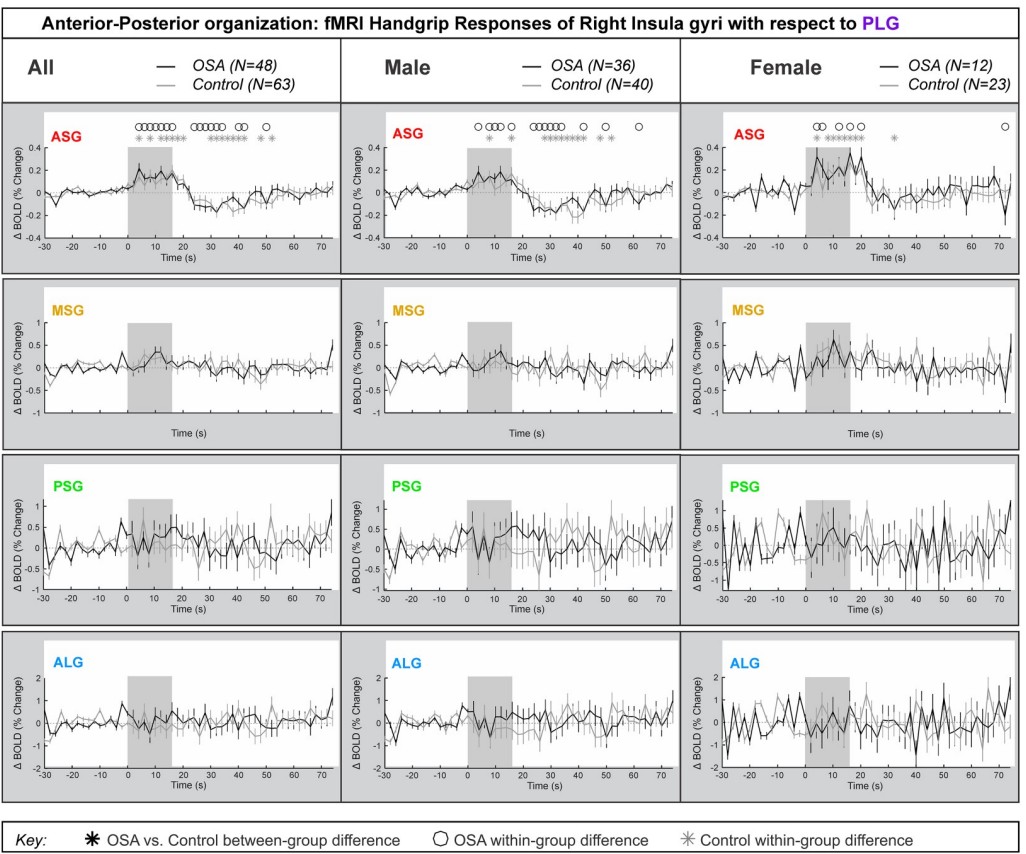

**Fig 4. Anterior-posterior organization of left insula.** Left hemisphere fMRI signals relative to PLG such that positive change reflects anterior dominance. Baseline (group mean ± SE), averaged over challenges, with time-points of significant increase or decrease relative to baseline within-group, and time-points of between-group differences (RMANOVA, p < 0.05) for the anterior-posterior responses by the left insular gyri in participants (all in left, male in middle and females in right columns respectively).

Consistent with prior studies in healthy people [46], we showed functional anterior-posterior and right-left organization of functional responses of gyri within the insula to a handgrip autonomic challenge. We observed minimal differences between OSA and control participants, suggesting that previously-described insular dysfunction occurs across the whole structure [15, 24–27]. Sex differences were apparent as previously shown, with anterior-most gyri exhibiting enhanced right-sided activation in females, and greater left-side activation in males [47]. We did not see substantial differences between OSA and controls in either combined-sex or separate male and female analyses. Age and resting HR showed associations with the magnitude of fMRI response to handgrip, but these associations did not change the patterns of gyral organization in OSA over control.

## Discussion

Increased anterior insular activity during the handgrip occurs in OSA and healthy groups, consistent with the established autonomic function of this sub-region found in both human neuroimaging and animal lesion studies [46, 48]. The absence of substantial OSA-related differences within the insula occurred despite other studies showing functional variations in

**Table 4. Left insula anterior-posterior fMRI organization with respect to PLG.**

| | Model details | All | | | | Male | | | | Female | | | |
|---|---|---|---|---|---|---|---|---|---|---|---|---|---|
| | | ASG | MSG | PSG | ALG | ASG | MSG | PSG | ALG | ASG | MSG | PSG | ALG |
| Main model: group, time | ChiSq (p value) | 72.54 (<0.001) | 66.82 (<0.001) | 52 (<0.001) | 55.07 (<0.001) | 68.07 (<0.001) | 56.94 (<0.001) | 39.3 (<0.001) | 43.18 (<0.001) | 12.98 (<0.01) | 13.46 (<0.01) | 16.13 (0.001) | 15 (0.002) |
| | Fit (−2 log-likelihood) | 13329.1 | 21139.5 | 26125.2 | 26548.5 | 9352.3 | 14636.2 | 18108.8 | 18403.8 | 3973.1 | 6411.7 | 7852.4 | 7971 |
| | Group effect p-value (mean over entire series for each group) | 0.83 | 0.72 | 0.97 | 0.73 | 0.92 | 0.92 | 0.88 | 0.76 | 0.85 | 0.33 | 0.62 | 0.89 |
| Handgrip response: within-group | Time (within-group effect of time) p-values | | | | | | | | | | | | |
| | **Main** | **<0.001** | **0.35** | **0.81** | **0.81** | **<0.001** | **0.32** | **0.77** | **0.68** | **0.054** | **0.57** | **0.77** | **0.78** |
| | Age | <0.001 | 0.35 | 0.81 | 0.81 | <0.001 | 0.32 | 0.77 | 0.68 | 0.054 | 0.57 | 0.77 | 0.78 |
| | Age x Group | <0.001 | 0.35 | 0.81 | 0.81 | <0.001 | 0.32 | 0.77 | 0.68 | 0.054 | 0.57 | 0.77 | 0.78 |
| | Age x Time | 0.26 | 0.55 | 0.55 | 0.26 | 0.47 | 0.36 | 0.16 | 0.20 | 0.50 | 0.44 | 0.38 | 0.30 |
| | Age x Group x Time | 0.26 | 0.55 | 0.55 | 0.26 | 0.47 | 0.36 | 0.16 | 0.20 | 0.50 | 0.44 | 0.38 | 0.30 |
| | HR | <0.001 | 0.35 | 0.81 | 0.81 | <0.001 | 0.32 | 0.77 | 0.68 | 0.054 | 0.57 | 0.77 | 0.78 |
| | HR x Group | <0.001 | 0.35 | 0.81 | 0.81 | <0.001 | 0.32 | 0.77 | 0.68 | 0.054 | 0.57 | 0.77 | 0.78 |
| | HR x Time | 0.68 | 0.31 | 0.36 | 0.40 | 0.83 | 0.23 | 0.38 | 0.38 | 0.32 | 0.81 | 0.84 | 0.78 |
| | HR x Group x Time | 0.68 | 0.31 | 0.36 | 0.40 | 0.83 | 0.23 | 0.38 | 0.38 | 0.32 | 0.81 | 0.84 | 0.78 |
| Handgrip response: between-group | Time X Group (between-group effect of time) p-values | | | | | | | | | | | | |
| | **Main** | **0.81** | **0.40** | **0.60** | **0.60** | **0.97** | **0.32** | **0.80** | **0.84** | **0.44** | **0.91** | **0.68** | **0.57** |
| | Age | 0.81 | 0.40 | 0.60 | 0.60 | 0.97 | 0.32 | 0.80 | 0.84 | 0.44 | 0.91 | 0.68 | 0.57 |
| | Age x Group | 0.81 | 0.40 | 0.60 | 0.60 | 0.97 | 0.32 | 0.80 | 0.84 | 0.44 | 0.91 | 0.68 | 0.57 |
| | Age x Time | 0.57 | 0.38 | 0.38 | 0.57 | 0.97 | 0.35 | 0.80 | 0.84 | 0.42 | 0.93 | 0.69 | 0.58 |
| | Age x Group x Time | 0.57 | 0.38 | 0.38 | 0.57 | 0.97 | 0.35 | 0.80 | 0.84 | 0.42 | 0.93 | 0.69 | 0.58 |
| | HR | 0.81 | 0.40 | 0.60 | 0.60 | 0.97 | 0.32 | 0.80 | 0.84 | 0.44 | 0.91 | 0.68 | 0.57 |
| | HR x Group | 0.81 | 0.40 | 0.60 | 0.60 | 0.97 | 0.32 | 0.80 | 0.84 | 0.44 | 0.91 | 0.68 | 0.57 |
| | HR x Time | 0.90 | 0.57 | 0.73 | 0.73 | 0.99 | 0.59 | 0.95 | 0.96 | 0.21 | 0.82 | 0.42 | 0.27 |
| | HR x Group x Time | 0.90 | 0.57 | 0.73 | 0.73 | 0.99 | 0.59 | 0.95 | 0.96 | 0.21 | 0.82 | 0.42 | 0.27 |

Salient statistics and p-values from 9 RMANOVA models for left insula anterior-posterior organization in three sets, namely "all" (both sexes), female and male. Full data are available online [59]. The main model (bold) is the interaction of group-by-time (fMRI = group + time + group x time), and measures of significance and fit are in the top rows of the table. The "Group" effect is the mean over the entire series and does not represent responses, and is not discussed. The two effects of interest "Epoch", which represents within-group responses over time, and "Epoch x Group", which represents between-group differences in responses. The p-values for these effects are shown for the 9 models. All models include the main effects plus additional mean or interaction terms. All interaction models also include means. For example, "Age x Time" is fMRI = group + time + group x time + age + age x time.

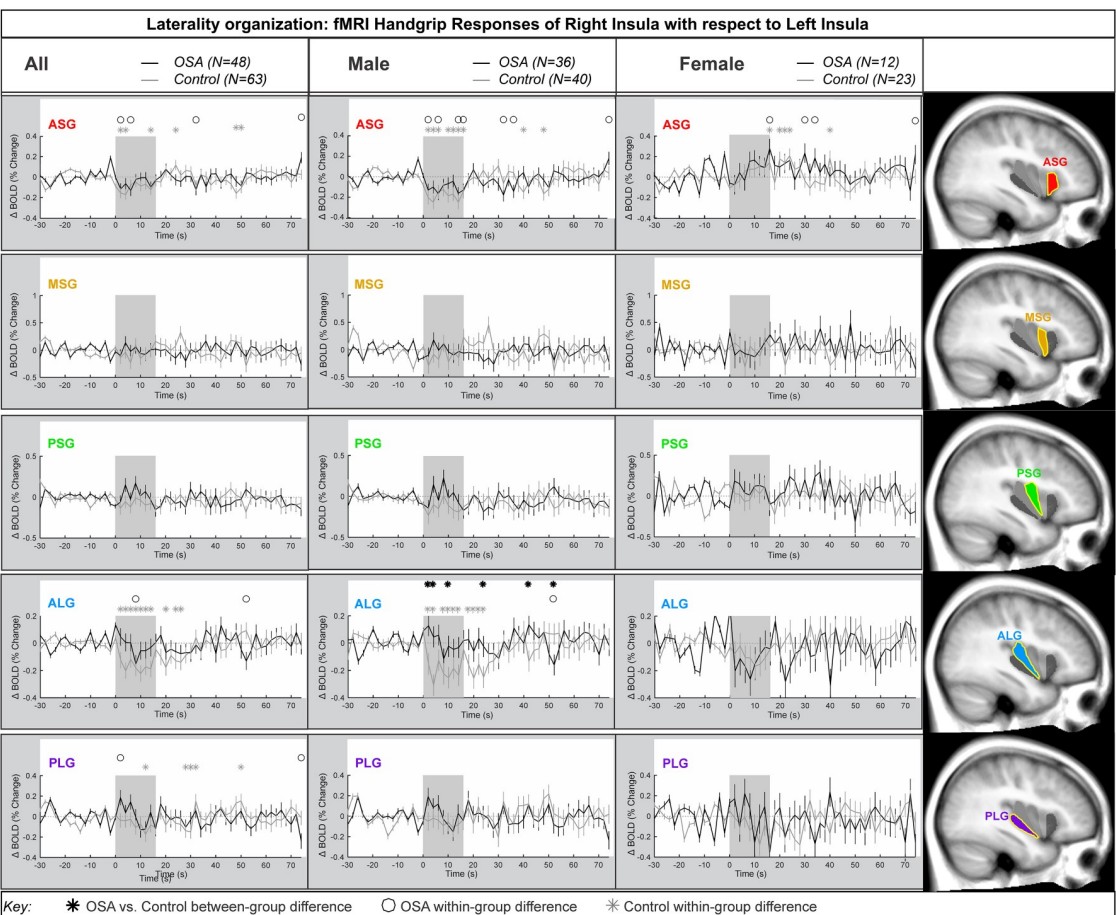

**Fig 5. Anterior-posterior organization of right insula.** Right hemisphere fMRI signals relative to PLG such that positive change reflects anterior dominance. Baseline (group mean ± SE), averaged over challenges, with time-points of significant increase or decrease relative to baseline within-group, and time-points of between-group differences (RMANOVA, p < 0.05) for the anterior-posterior responses by the right insular gyri in participants (all in left, male in middle and females in right columns respectively).

networks containing the structure [32, 33, 38]. Together with existing evidence, the present findings therefore suggest that within-structure organization of the insula is intact in OSA, but function of the structure as a whole is impaired.

The insula plays a key role in cardiovascular regulation, and hence is relevant to this main comorbidity of OSA. Primate studies have revealed that the anterior, agranular insula receives inputs from brainstem regions such as the nucleus of the solitary tract (NTS) and sends direct projections to caudal regions including the midbrain periaqueductal gray, parabrachial nucleus and NTS [60]. A motor command alone can evoke increased heart rate and blood pressure [61], even if the target muscle is paralyzed [62, 63], and animal electrophysiological studies have revealed that unmyelinated and small-diameter myelinated nerve fibers are responsible for muscle contraction-evoked cardiovascular and respiratory responses [64, 65]. Furthermore, these muscle afferents project to the premotor sympathetic neurons in the rostral ventrolateral medulla (RVLM) via the NTS [66], and healthy individuals show increased signal intensity changes in the parabrachial nucleus, NTS and RVLM during static handgrip [67]. As discussed previously [68], the insula modulates autonomic function via projections to autonomic outflow regions in the brainstem, often indirectly via the hypothalamus [21]. The

**Table 5. Right insula anterior-posterior fMRI organization with respect to PLG.**

| | Model details | All | | | | Male | | | | Female | | | |
|---|---|---|---|---|---|---|---|---|---|---|---|---|---|
| | | ASG | MSG | PSG | ALG | ASG | MSG | PSG | ALG | ASG | MSG | PSG | ALG |
| Main model: group, time | ChiSq (*p* value) | 205.5 (<0.001) | 76.74 (<0.001) | 58.29 (<0.001) | 58.18 (<0.001) | 175.52 (<0.001) | 59.34 (<0.001) | 48.71 (<0.001) | 48.1 (<0.001) | 29.72 (<0.001) | 29.33 <0.001) | 15.75 (0.001) | 16.9 (<0.001) |
| | Fit (−2 log-likelihood) | 3633.6 | 15660 | 24697.8 | 28268.3 | 2740.1 | 10938.2 | 17143.6 | 19610.7 | 1007.2 | 4661.5 | 7392.7 | 8443.7 |
| | Group effect p-value (mean over entire series for each group) | 0.40 | 0.72 | 0.76 | 0.55 | 0.47 | 0.72 | 0.61 | 0.44 | 0.25 | 0.20 | 0.68 | 0.67 |
| Handgrip response: within-group | Time (within-group effect of time) p-values | | | | | | | | | | | | |
| | **Main** | *<0.001* | *<0.001* | **0.64** | **0.84** | *<0.001* | *0.016* | **0.59** | **0.59** | *<0.001* | *<0.001* | **0.45** | **0.76** |
| | Age | <0.001 | <0.001 | 0.64 | 0.84 | <0.001 | 0.016 | 0.59 | 0.59 | <0.001 | <0.001 | 0.45 | 0.76 |
| | Age x Group | <0.001 | <0.001 | 0.64 | 0.84 | <0.001 | 0.016 | 0.59 | 0.59 | <0.001 | <0.001 | 0.45 | 0.76 |
| | Age x Time | *0.02* | 0.43 | 0.20 | 0.35 | 0.14 | 0.56 | 0.14 | 0.32 | *0.006* | 0.43 | 0.32 | 0.25 |
| | Age x Group x Time | *0.02* | 0.43 | 0.20 | 0.35 | 0.14 | 0.56 | 0.14 | 0.32 | *0.006* | 0.43 | 0.32 | 0.25 |
| | HR | <0.001 | <0.001 | 0.64 | 0.84 | <0.001 | 0.016 | 0.59 | 0.59 | <0.001 | <0.001 | 0.45 | 0.76 |
| | HR x Group | <0.001 | <0.001 | 0.64 | 0.84 | <0.001 | 0.016 | 0.59 | 0.59 | <0.001 | <0.001 | 0.45 | 0.76 |
| | HR x Time | 0.85 | 0.62 | 0.84 | 0.73 | 0.84 | 0.73 | 0.73 | 0.72 | 0.73 | 0.17 | 0.54 | 0.75 |
| | HR x Group x Time | 0.85 | 0.62 | 0.84 | 0.73 | 0.84 | 0.73 | 0.73 | 0.72 | 0.73 | 0.17 | 0.54 | 0.75 |
| Handgrip response: between-group | Time X Group (between-group effect of time) p-values | | | | | | | | | | | | |
| | **Main** | **0.85** | **0.57** | **0.69** | **0.67** | **0.80** | **0.51** | **0.80** | **0.90** | **0.92** | **0.71** | **0.46** | **0.36** |
| | Age | 0.85 | 0.57 | 0.69 | 0.67 | 0.80 | 0.51 | 0.80 | 0.90 | 0.92 | 0.71 | 0.46 | 0.36 |
| | Age x Group | 0.85 | 0.57 | 0.69 | 0.67 | 0.80 | 0.51 | 0.80 | 0.90 | 0.92 | 0.71 | 0.46 | 0.36 |
| | Age x Time | 0.85 | 0.55 | 0.66 | 0.65 | 0.78 | 0.54 | 0.81 | 0.91 | 0.86 | 0.75 | 0.47 | 0.35 |
| | Age x Group x Time | 0.85 | 0.55 | 0.66 | 0.65 | 0.78 | 0.54 | 0.81 | 0.91 | 0.86 | 0.75 | 0.47 | 0.35 |
| | HR | 0.85 | 0.57 | 0.69 | 0.67 | 0.80 | 0.51 | 0.80 | 0.90 | 0.92 | 0.71 | 0.46 | 0.36 |
| | HR x Group | 0.85 | 0.57 | 0.69 | 0.67 | 0.80 | 0.51 | 0.80 | 0.90 | 0.92 | 0.71 | 0.46 | 0.36 |
| | HR x Time | 0.92 | 0.74 | 0.81 | 0.78 | 0.94 | 0.77 | 0.95 | 0.98 | 0.73 | 0.52 | 0.20 | 0.13 |

Salient statistics and p-values from 9 RMANOVA models for right insula anterior-posterior organization in three sets (mixed, male, female). Full data are available online [59]. The main model (bold) is the interaction of group-by-time (fMRI = group + time + group x time), and statics of significance and fit are in the top rows of the table. The "Group" effect is the mean over the entire series and does not represent responses, and is not discussed. The two effects of interest "Time", which represents within-group responses over time, and "Time x Group", which represents between-group differences in responses. The p-values for these effects are shown for the 9 models. All models include the main effects plus additional mean or interaction terms. All interaction models also include means. For example, "Age x Time" is fMRI = group + time + group x time + age + age x time.

hypothalamus, in particular, is tightly linked with sympathetic activity and has direct projections to and from the insula [19, 69–72]. This close connection pattern is consistent with resting state neuroimaging studies that revealed altered baseline function in insular cortices and altered functional connectivity from the insula to other autonomic-related brain regions in OSA [25, 26], so a next area of investigation might be studies of functional interactions between insular, hypothalamic and brainstem activity during autonomic responses.

The magnitude of the fMRI responses to the task and of group differences was in the order of 0.3%, as seen in the time trends. For neuroimaging studies generally, a change of 1% in the BOLD signal is considered close to maximal, and the effects seen here are consistent with moderate effect sizes [73, 74]. Additionally, some effect sizes were minimal as the hypothesized differences were not observed; these negative findings were included in the results.

While a role for the brainstem in generating cardiovascular changes to muscle contractions is clear, cardiovascular changes can also occur during imagination of muscle contraction, that is without peripheral input [75]. The insular cortex and anterior cingulate cortex (ACC) have been implicated in integrating motor command and cardiovascular changes during exercise, which is supported by numerous brain imaging investigations [67, 75–77]. The anterior insular also receives inputs from and projects to the ACC, and cardiovascular changes during imagined movement are apparently associated with activity changes in the ACC and insular cortex [75]. The findings from previous studies are consistent with the data presented here; that is, during a handgrip challenge, the increase in HR is associated with signal intensity changes within the anterior agranular insula and not in other more-posterior regions of the structure.

While only limited evidence exists on neural function-related sex differences in OSA, here, we found that both OSA and control females showed higher anterior signal dominance in both left and right insulae during an autonomic challenge. However, relative to control participants, OSA males showed left sided anterior dominance. This influence did not appear to be modulated by resting HR. However, resting HR was high in male, but not female in OSA relative to healthy groups, which could reflect different resting sympathetic tone, and potentially contribute to a sex-specific ceiling effect in fMRI responses. Given previous resting-state and neurotransmitter findings of the insula in OSA [25, 26, 35–37], the question arises for future studies whether the baseline neural state is altered in a different manner in OSA females and males. Estrogen exerts sympathetic influences on the insula [78]. Furthermore, these effects appear to be mediated by GABA, and given the lower GABA in OSA, complex interactions between estrogen and OSA-related GABA reductions affecting autonomic regulation by the insula could arise. This left-sided parasympathetic and right-sided sympathetic insular laterality is more a bias than an absolute distinction between sympathetic and parasympathetic activity, since there are common cardiovascular responses to stimulation across multiple insular regions [50, 79]. Oppenheimer and colleagues [51] showed lateralization in insular cortex stimulation-elicited differential cardiovascular rhythm changes in epileptic patients, with right insula stimulation triggering sympathetic and left insula, parasympathetic effects. Removal of the right insula in rats leads to increased parasympathetic activity [80]. Under these assumptions, the findings suggest that males showed more parasympathetic withdrawal and females showed more sympathetic activation during the handgrip task, consistent with previous findings [47]. Consistent with our study in healthy individuals [46], here we report that the direction of left-right organization is similar in both OSA and control groups, with higher activity on the right side only for the females and not males.

Interpretation of these findings is limited by several factors. The handgrip task was short, and likely did not elicit metabolic effects or the increase in sympathetic activity found in healthy people. The grip strength was a subjective rating, but a percentage of maximum is the

standard approach for handgrip tasks; this issue may have added variability to the data. The sample was originally powered for OSA-control differences, but not sex-specific effects; nevertheless, since we know there are sex differences in OSA and healthy cardiovascular and neural function, we decided to provide sex-specific results. Another limitation due to the original study intent is the lack of information in females on menstrual cycle, menopausal status, or use of hormones; these factors are all associated with autonomic influences [81, 82]. Sleep study parameters were based on the sleep study reports provided, and did not typically include quantification of awakenings, sleep onset latency, sleep efficiency or other parameters. Sleep quality in particular has been associated with changes in insular function [83]. With respect to neuroimaging, insula cortex folding is not constant across people, and the gyri are likely accurate only to within a few mm; fMRI signals are also limited in spatial accuracy due to the diffuse nature of the BOLD effect. Finally, representation of the OSA group to the broader patient population is limited.

In conclusion, functional response organization of gyri within the insular cortex is not substantially altered during a handgrip challenge in OSA subjects. This finding is similar to the lack of substantial OSA differences in those gyri in response to a Valsalva maneuver, and suggests that the anterior and right dominance of responses within the insula to autonomic stimuli may remain largely intact in the condition. It appears that in response to the motor driven handgrip task the autonomic functional organization of insular gyri appear muted and only the central command motor task aspect of the insula appears to differ in male OSA subjects compared to controls. Females showed higher anterior and right fMRI signal dominance in insula gyri compared with males, but the sample was insufficiently large to generalize with confidence. Since central autonomic regulation is impaired in OSA, given the peripheral weakened responses, questions remain regarding the resting state functional activity and connectivity with other autonomic regions such as the hypothalamus and brainstem.

## Supporting information

**S1 File. Contains phone screening questions and topics for in-person semi-structured interview to assess potential undiagnosed OSA in control participants.**
(DOCX)

**S2 File. Contains masks of the 10 gyral parcellations (five gyri in two hemispheres) in nifti format.** The masks are in template space. ASG: anterior short gyrus; MSG: mid short gyrus; PSG: posterior short gyrus; ALG: anterior long gyrus; PLG: posterior long gyrus.
(ZIP)

## Author Contributions

**Conceptualization:** Rajesh Kumar, Luke A. Henderson, Ronald M. Harper.

**Data curation:** Amrita Pal, Jennifer A. Ogren, Ravi S. Aysola, Rajesh Kumar, Luke A. Henderson, Ronald M. Harper, Paul M. Macey.

**Formal analysis:** Amrita Pal, Ravi S. Aysola, Paul M. Macey.

**Investigation:** Jennifer A. Ogren, Luke A. Henderson, Ronald M. Harper, Paul M. Macey.

**Methodology:** Amrita Pal, Jennifer A. Ogren, Paul M. Macey.

**Project administration:** Paul M. Macey.

**Resources:** Paul M. Macey.

**Software:** Paul M. Macey.

**Supervision:** Paul M. Macey.

**Validation:** Amrita Pal, Ronald M. Harper, Paul M. Macey.

**Visualization:** Amrita Pal, Paul M. Macey.

**Writing – original draft:** Amrita Pal.

**Writing – review & editing:** Luke A. Henderson, Ronald M. Harper, Paul M. Macey.

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
