## [Decision Letter · Decision Letter 0]

10 Dec 2020

PONE-D-20-32454

Insular Functional Organization during Handgrip in Obstructive Sleep Apnea Females and Males

PLOS ONE

Dear Dr. Macey,

Thank you for submitting your manuscript to PLOS ONE. After careful consideration, we feel that it has merit but does not fully meet PLOS ONE’s publication criteria as it currently stands. Therefore, we invite you to submit a revised version of the manuscript that addresses each the points raised during the review process.

You will find the comments from the two reviewers below. As a small note, there appears to be a typo in Comment #4 from Reviewer 2. Specifically, the sentence "Moreover mean saturation ( whether in the scanner or during sleep) is not really useful because can indicate only a dip in saturation" should reference minimum saturation as opposed to mean. 

We look forward to receiving your revised manuscript.

Kind regards,

Bradley R. King

Academic Editor

PLOS ONE

Journal Requirements:

2)  Thank you for stating the following in the Acknowledgments Section of your manuscript:

[This research was supported by the National Institute of Nursing Research NR-017435 and National Institute of Heart, Lung and Blood Institute HL135562. The funders had no role in study design, data collection and analysis, decision to publish, or preparation of the manuscript.]

 [The funders had no role in study design, data collection and analysis, decision to

publish, or preparation of the manuscript.]

3) Please provide additional details regarding participant consent. In the ethics statement in the Methods and online submission information, please ensure that you have specified whether consent was informed."

4)  In your Methods section, please provide additional information about the participant recruitment method and the demographic details of your participants. Please ensure you have provided sufficient details to replicate the analyses such as:

a) the recruitment date range (month and year),

b) a description of any inclusion/exclusion criteria that were applied to participant recruitment,

c) a statement as to whether your sample can be considered representative of a larger population, and

d) a description of how participants were recruited."

5)  Please include additional information regarding the interview guide used in the study and ensure that you have provided sufficient details that others could replicate the analyses. For instance, if you developed it as part of this study and it is not under a copyright more restrictive than CC-BY, please include a copy, in both the original language and English, as Supporting Information.

Reviewers' comments:

Reviewer's Responses to Questions

**Comments to the Author**

1. Is the manuscript technically sound, and do the data support the conclusions?

Reviewer #1: Yes

Reviewer #2: Partly

2. Has the statistical analysis been performed appropriately and rigorously? 

Reviewer #1: Yes

Reviewer #2: Yes

3. Have the authors made all data underlying the findings in their manuscript fully available?

Reviewer #1: Yes

Reviewer #2: Yes

4. Is the manuscript presented in an intelligible fashion and written in standard English?

Reviewer #1: Yes

Reviewer #2: Yes

5. Review Comments to the Author

Reviewer #1: This is an fMRI study with 111 subjects. Results showed that insular gyri functional responses to handgrip differ in OSA vs controls in a sex-based manner, but only in laterality of one gyrus.

The Introduction should be further developed to explain to readers about the relevance of handgrip for OSA patients. What is the point of handgrip? By reading through the current version of the Introduction, it is difficult for the reviewer to understand the connection or relevance between the two. Afterwards the text should explain why sexual differences in cerebral processing of handgrip in OSA patients are important. Why not foot grip or intentional forceful exhalation?

From the Figures reporting fMRI time trend results, it seems that the differences were very subtle. Please further explain to the readers the significance of differences with such a small magnitude.

There are multiple neuroimaging meta-analysis of OSA patients in terms of morphometric, functional connectivity and activation changes (c.f. 10.1016/j.neubiorev.2016.03.026 ; 10.1111/jsr.12857 ; 10.2147/NDT.S161085 ; just to name a few as examples). Did the current results show spatial convergence with brain regions reported in previous meta-analysis? The Discussion is infertile without references to and comparison with the multiple meta-analysis.

Reviewer #2: In this article, the authors sought to investigate the neural underpinning of an hand grip challenge in individuals with OSA. They specifically focus on BOLD signal change in different insular giri based on previous literature. The authors also look at sex based differences although this seems to be an a-posteriori analysis. They found no differences in OSA and controls within females but a greater activation of the right anterior ASG as compared to males. They found differences between OSA and controls in males with OSA males having lower activation of the left ALG.

The paper is overall interesting, however I have a few comments:

1) The sex based analysis seems to be a second thought as expressed by the authors in the discussion. This is evident from the fact that in the methods it is not clear how the authors will look at sex differences. In the introduction (line 76) the authors state that they will control for sex differences. This means looking at the brain activations above and beyond the effects of sex. However the goal in the paper was to look at different mechanisms in males and females which would be better done with stratified analysis, which I believe the authors actually did. In light of this I suggest to re write the last paragraph of the introduction an methods to clearly state that the main interest is Sex differences and that separate analyses in males and females will be conducted to assess them.

2) Furthermore, in the introduction the authors explain how sex differences may be due to hormonal effects that impact both susceptibility to OSA and sympathetic responses. Did the authors collect information about menstrual cycle, menopause ( average age for females in 50), use of hormonal replacement therapy? If this information was not collected this should be discussed as a limitation.

3) More details are needed about how these participants were diagnosed for OSA. Did they undergo a full level 2 polysomnography or just an apnea test? Moreover more information about sleep quality and other sleep parameters should be reported ( i.e. sleep efficiency, awakenings etc.). The authors mention a semi structured interview to assess symptoms of OSA. This information should be reported.

4) In table 1, the authors report AHI values that I believe were assessed by one of the aforementioned methods but then report the oxygen saturation collected in the scanner. This is unclear. Please report oxygen desaturation index and mean saturation through the sleep study night. Moreover mean saturation ( whether in the scanner or during sleep) is not really useful because can indicate only a dip in saturation. Mean saturation would be more useful.

5) What is the reason to use VBM 8 based on DARTEL ? Where you expecting structural abnormalities, atrophy in these patients. Please specify.

6) Please report MNI coordinates for the different ROIs.

7) Previous studies reported differential activation of the insula according to individuals sleep quality (Guadagni et al. 2018, https://doi.org/10.1111/ejn.14124). Sleep quality should be used as a covariate in the models or at least this should be discussed as a possible confounder.

8) The discussion should be re organized. The findings should be discussed in the first paragraph then connected to existing literature.

Minor:

- Why is line 151 italiacized?

- Table 2: please report that the values are p values either in the legend. It took me a while to see it in the table which is perhaps a little busy.

6. PLOS authors have the option to publish the peer review history of their article (what does this mean?). If published, this will include your full peer review and any attached files.

Reviewer #1: No

Reviewer #2: **Yes: **Dr. Veronica Guadagni

---

## [Author Response · Author response to Decision Letter 0]

29 Dec 2020

Please see cover letter and response to reviewers file.

---

## [Decision Letter · Decision Letter 1]

19 Jan 2021

Insular Functional Organization during Handgrip in Females and Males with Obstructive Sleep Apnea

PONE-D-20-32454R1

Dear Dr. Macey,

We’re pleased to inform you that your manuscript has been judged scientifically suitable for publication and will be formally accepted for publication once it meets all outstanding technical requirements.

Kind regards,

Bradley R. King

Academic Editor

PLOS ONE

Additional Editor Comments (optional):

Reviewers' comments:

Reviewer's Responses to Questions

**Comments to the Author**

1. If the authors have adequately addressed your comments raised in a previous round of review and you feel that this manuscript is now acceptable for publication, you may indicate that here to bypass the “Comments to the Author” section, enter your conflict of interest statement in the “Confidential to Editor” section, and submit your "Accept" recommendation.

Reviewer #1: All comments have been addressed

Reviewer #2: All comments have been addressed

2. Is the manuscript technically sound, and do the data support the conclusions?

Reviewer #1: (No Response)

Reviewer #2: Yes

3. Has the statistical analysis been performed appropriately and rigorously? 

Reviewer #1: (No Response)

Reviewer #2: Yes

4. Have the authors made all data underlying the findings in their manuscript fully available?

Reviewer #1: (No Response)

Reviewer #2: Yes

5. Is the manuscript presented in an intelligible fashion and written in standard English?

Reviewer #1: (No Response)

Reviewer #2: Yes

6. Review Comments to the Author

Reviewer #1: (No Response)

Reviewer #2: (No Response)

7. PLOS authors have the option to publish the peer review history of their article (what does this mean?). If published, this will include your full peer review and any attached files.

Reviewer #1: No

Reviewer #2: No

---

## [Editor Report · Acceptance letter]

3 Feb 2021

PONE-D-20-32454R1 

Insular Functional Organization during Handgrip in Females and Males with Obstructive Sleep Apnea 

Dear Dr. Macey:

I'm pleased to inform you that your manuscript has been deemed suitable for publication in PLOS ONE. Congratulations! Your manuscript is now with our production department. 

Kind regards, 

on behalf of

Dr. Bradley R. King 

Academic Editor

PLOS ONE